# Role of Non-Coding RNAs in Acute Myeloid Leukemia

**DOI:** 10.3390/ncrna11050070

**Published:** 2025-09-19

**Authors:** Shailendra S. Maurya, Sarita Maurya, Sumit K. Chaturvedi

**Affiliations:** 1Chromatin Architecture Laboratory, Department of Hematology (SCRC), Sanjay Gandhi Postgraduate Institute of Medical Sciences, Lucknow 226014, India; 2Department of Biophysics, University of Delhi South Campus, New Delhi 110021, India

**Keywords:** non-coding RNA (ncRNAs), long non-coding RNA (lncRNA), circular (circ), acute myeloid leukemia (AML)

## Abstract

Acute myeloid leukemia (AML) is a highly heterogeneous disease, with significantly higher incidence and fatality rates in the elderly. Even with recent decades of research progress in AML, the exact etiology of this deadly disease is still not fully understood, with recent advancements in sequencing technologies highlighting the role of a growing number of non-coding RNAs (ncRNAs) that are intimately associated with AML leukemogenesis. These ncRNAs have been found to have a significant role in leukemia-related cellular processes such as cell division, proliferation, and death. A few of these non-coding RNAs exhibit potential as prognostic biomarkers. The three main groups of ncRNAs that contribute unique activities, especially in cancer, are microRNAs (miRNAs), long non-coding RNAs (lncRNAs), and circular RNAs (circRNAs). Their existence or altered expression levels frequently offer vital information on the diagnosis, course of treatment, and follow-up of cancer patients. The identification of ncRNAs has opened up new avenues for the diagnosis, prognosis, and therapy of acute myeloid leukemia. In order to provide a clear understanding of the significant influence that lncRNAs have on prognostic predictions and diagnostic accuracy in AML, this review aims to provide a comprehensive and insightful understanding of how these molecules actively participate in the complex landscape of the disease.

## 1. Introduction

AML has undergone significant changes over time. Previously considered to be a single type of acute leukemia, AML has transformed into a diverse group of subtypes, each with unique molecular and pathophysiologic characteristics [1]. Figure 1 shows how AML starts in the bone marrow niche. The use of tailored treatments has been made possible by the realization of this heterogeneity. AML is an aggressive blood cancer that occurs when the bone marrow’s immature myeloid cells begin to proliferate and differentiate improperly [1]. Healthcare professionals can now customize treatment plans to the unique features of each AML subtype rather than using a one-size-fits-all strategy. By minimizing potential side effects and consequences, this customized approach not only increases the overall effectiveness of therapies but also enhances the quality of life for patients. Even though there is an increasing number of therapy options available for AML, the illness’s management remains difficult. Unfortunately, even after reaching an initial remission, a large percentage of AML patients eventually relapse and die [2,3]. This fact highlights the ongoing need for better AML prognoses and therapy.

There are just a few AML cases where specific factors, including prior chemotherapy treatments or chemical exposure, have been identified. However, genetic abnormalities, such as single gene mutations or chromosomal abnormalities, are thought to be the cause of most AML cases. There is no apparent connection between the origins of these genetic changes and any extrinsic causal factors, which frequently serve as the leading causes of AML development [4]. The World Health Organization (WHO) divided AML into six categories in its 2016 guidelines, each with unique traits and factors to take into account. For researchers to comprehend the variety of AML and customize treatments to the unique features of each AML subtype, the WHO classification system is essential (Table 1).

## 2. Current Methods for Identification of ncRNAs

Numerous ncRNAs are produced and released by malignant cells and are found in biological fluids like urine or blood. Their existence or changed expression levels frequently offer vital information on the diagnosis, course of treatment, and follow-up of cancer patients [5]. The identification of ncRNAs has opened up new avenues for the diagnosis, prognosis, and therapy of AML. In order to provide a clear understanding of the significant influence that miRNAs, circRNAs, and lncRNAs have on prognostic predictions and diagnostic accuracy in AML, this review aims to provide a comprehensive and insightful account of how these molecules actively participate in the complex landscape of the disease.

Currently, RNA sequencing, microarrays, and qRT-PCR are the primary techniques used to identify and measure ncRNA biomarkers in biopsies. However, until these methods can be easily incorporated into clinical practice, they frequently need to undergo significant optimization. Therefore, the development of a quick, standardized, and clinically useful technique is urgently needed in order to help move ncRNA profiling from bench-top research to real-world bedside applications. Bloomfield et al. developed a probe assay based on the nCounter platform (*nSolver Analysis Software, version 4.0*) to meet this demand [6]. The goal of this novel strategy is to quantify prognostic lncRNAs with accuracy. Notably, the nCounter platform’s viability for clinical use has already been demonstrated by its use in an FDA-approved test for measuring RNA expression in breast cancer patients [7]. Focusing on oncogenic lncRNAs is an appealing strategy in the search for efficient cancer treatments [8]. The benefits of synthetic oligonucleotide-based molecular products—such as their simplicity of dosage control, minimal immunogenicity, and lack of dangers associated with genome integration—have led to their widespread exploration as a technology [9]. An essential component of this strategy’s success is comprehending where lncRNAs are located within cells. This information is essential for choosing the best method to modify lncRNAs effectively; siRNAs could be a helpful instrument in this regard. By enlisting the RNA-induced silencing complex (RICS), they are intended to be complementary and antisense to the target lncRNAs, causing target degradation. Although they are rather effective against cytoplasmic targets, they are not always successful in targeting nuclear lncRNAs. While siRNAs have not yet been suggested as a therapy for AML, phase I/II trials have been conducted for various diseases [10]. One prominent example is the reduction in endometrial cancer progression in vivo by siRNA targeting the lncRNA HOTAIR [11]. This discovery emphasizes the possibility of using lncRNA HOTAIR targeting as a novel therapeutic approach for endometrial cancer. The same siRNA approach may be taken into consideration in the management of AML due to the upregulation of HOTAIR in this disease, creating new opportunities for therapies in this setting. It is possible to target lncRNAs linked to chemoresistance utilizing interference methods, thereby improving medication response and treatment outcomes for AML. For this reason, a large number of biotech companies are actively working on developing oligonucleotide-based solutions [12]. However, improving delivery efficiency and ensuring patients achieve long-lasting results are significant challenges. Stability and interferon induction problems can be addressed by modifying ASOs and small interfering RNAs (siRNAs) [8]. The use of ncRNAs as therapeutic agents in clinical practice is still in its infancy due to the scarcity of AML trials. AML’s intrinsic biological complexity and non-standardized sample collection and quantification methods are probably to blame for the lack of reproducible results across cohorts. It is essential to carry out carefully planned cohort studies with sufficient sample sizes and validate the findings in separate cohorts in order to demonstrate the clinical value of ncRNAs. In addition, determining which are the most promising ncRNA targets and creating secure and efficient ncRNA therapeutic instruments depend on the functional and molecular characterization of ncRNAs. Although our comprehension of this complexity has advanced, it remains challenging to apply this knowledge in a therapeutic setting.

## 3. Role of lncRNAs and Cancer

### 3.1. Overview of lncRNAs and Their Classification

Only three percent of the human genome ultimately produces mRNAs, which code for proteins. It is well established that merely 2% of the human genome is responsible for encoding proteins. In contrast, approximately 70–90% undergo transcription, producing an enormous number of transcripts that belong to the non-coding part of the genome [13,14]. These transcripts have a significant proportion of lncRNAs that are drawing considerable interest as novel regulators of RNAs in various physiological processes [15,16,17]. The extensive role of ncRNAs, which are classified according to their length, structure, and chromosomal position, is shown by this contrast. LncRNAs are classified based on their size. Those with fewer than 200 nucleotides are categorized as short ncRNAs, while those with more than 200 nucleotides are classified as lncRNAs [18]. Further, lncRNAs can be classified based on their location relative to the protein-coding region; these categorizations include intronic, sense, antisense, and bidirectional types of lncRNA [19]. Apart from this, there are four other types of ncRNAs that contribute unique activities, especially in cancer: miRNAs, lncRNAs, circRNAs, and PIWI-interacting RNAs (piRNAs). miRNAs are a family of small RNAs that are around 22 nucleotides long and bind to complementary target mRNA. miRNAs also can bind other RNA; apart from this, lncRNAs frequently act as sponges for miRNAs. Thus, they play an important regulatory role. The targeted mRNA is degraded as a result of the binding event that initiates the RISC assembly [20]. On the other hand, circRNAs and lncRNAs are longer than 200 nucleotides. Whereas circRNAs assume a circular, ring-like structure, lncRNAs take on a linear form. Introns, exons, and 5′ and 3′ untranslated regions are some of the chromosomal locations from which either of these ncRNAs can originate [21]. Their intricate secondary structures enable interactions with proteins, RNA, and DNA, providing a wide range of regulatory roles in gene expression and cellular processes. Strong evidence highlights the critical roles ncRNAs play in human cancers [22]. Emerging evidence has indicated that lncRNAs function as important regulators in cancer, affecting proliferation, differentiation, therapy resistance, and tumor suppression and serving as critical biomarkers [19]. These molecules can act either as tumor suppressors, inhibiting cancer initiation/progression, or as oncogenes, influencing tumor development. In their oncogenic capacity, they enhance cancer cell survival, growth, and migration and contribute to the development of therapy resistance [23].

### 3.2. Molecular Function of lncRNAs

Uncontrolled proliferation of myeloid progenitors is a common phenomenon in acute myeloid leukemia that creates a block in the differentiation stages. The hallmark of AML disease is the differentiation block, leading to the accumulation of immature myeloblasts [19]. Among the various long non-coding RNAs playing a role in this process, HOTTIP has recently been recognized for its role in inhibiting the differentiation of myeloid cells in AML [24]. Another potential lncRNA has recently emerged as a crucial regulator of myeloid cell differentiation in NPM1-mutant AML [25]. One lncRNA related to NPM1 mutant AML is LONA, which displays higher expression in NPM1 mutant AML compared to NPM1 wild-type AML [25].

Additionally, the nuclear localization of LONA exhibits an inverse correlation with NPM1 protein localization [25]. Another lncRNA, HOTAIR, that is usually associated with solid tumors, has a dual role by triggering differentiation events and enhancing the self-renewal capacity [26]. Lastly, HOXA10-as, another lncRNA, has been identified as a contributor to the disruption of myeloid cell differentiation [27].

LncRNA also plays a role in developing drug resistance; for example, the lncRNA DANCER has been recognized to confer resistance to the conventional drug regimen cytarabine (Ara-C) in AML cell lines [28]. Likewise, the lncRNA UCA1 has been recognized as a mediator of chemotherapy resistance via the process of miRNA sponging [29].

The lncRNA XIST, previously identified as an initial long non-coding RNA, demonstrated as a miR-29a sponge and thus modulated the suppression of the oncogene MYC [30]. Another example is the lncRNA HOTAIRM1 (HOTAIR), which is notably specific to myeloid cells. HOTAIRM1 expression has been associated with the differentiation of myeloid cells, suggesting a direct role in the progression of AML [31]. In a recent study by Cui et al., LINC00152 was identified as a specific lncRNA for CD34+CD38 leukemia stem cells (LSC), and they also reported that its expression showed a strong correlation with the “LSC17” gene expression signature [32,33], indicating a direct involvement of LINC00152 in leukemia stemness. They further reported that downregulating LINC00152 enhances LSC sensitivity to DOX and leads to a decrease in the expression of the DNA damage repair protein PARP1 [32].

On the other hand, ncRNAs that exhibit tumor-suppressive characteristics serve as protectors of genomic stability by impeding excessive cell division and facilitating the restoration of DNA damage [33].

LncRNA NR-104098 is downregulated in AML cells following ATRA-induced differentiation [34]. LOUP is another lncRNA that promotes myeloid differentiation [35]. LSC self-renewal is a primary event that induces AML initiation. LncRNAs that regulate LSC self-renewal play a crucial role in inhibiting the disease. For example, the lncRNA MAGIC2AS3 is observed to inhibit the LSC self-renewal capacity and function as a tumor suppressor in AML [36]. Another lncRNA, NEAT1, has been reported as a tumor suppressor in AML [37]. A recent study demonstrated that MEG3 expression is decreased in AML patient samples compared to healthy CD34+ cells [37].

Increased expression of lncRNA has been associated with an adverse prognosis in AML. PANDAR exhibits increased expression levels in AML patients compared to healthy controls. ROC analysis demonstrated that the PANDAR expression level could serve as a diagnostic biomarker, which can distinguish AML patients from healthy controls [38]. Likewise, FBXLI9-AS1 is a potential diagnostic marker for AML patients [39]. Moreover, the lncRNA LINC00899 was reported as being upregulated in the serum and bone marrow of AML patients compared to healthy controls, showing capabilities for use as a potential biomarker [40].

Apart from this, researchers also methodically found lncRNAs in specific leukemia variations, including childhood MLL-rearranged acute lymphoblastic leukemia (ALL), and demonstrated their association with many target protein-coding genes [41]. This finding implies that these lncRNAs regulate the expression of genes and may therefore have an effect on the development and progression of leukemia [42]. Two lncRNAs have been identified as critical moderators of erythro-megakaryocytic development in the setting of acute megakaryoblastic leukemia [43]. This suggests that these lncRNAs have a direct role in controlling the processes of blood cell maturation and differentiation, especially those related to the production of megakaryocytes and red blood cells. Moreover, these lncRNAs are associated with the continuous expansion of leukemic cells associated with this subtype of leukemia. Leukemic cells continue to proliferate due to their existence and functional activity, which highlights the possible significance of lncRNAs in influencing the genesis and course of particular types of leukemia [42]. A thorough comprehension of these lncRNAs’ functions in leukemia may provide important insights into the underlying molecular pathways and may identify candidates for therapeutic targeting.

LncRNAs are essential for controlling multiple aspects of AML cell activity, such as proliferation regulation, cell cycle regulation, and apoptosis initiation. One such instance is the lncRNA PVT1 [44]. As a result of the amplification of the genomic area 8q24.21, which is positioned next to the MYC gene, PVT1′s genomic location functions in conjunction with MYC in about 10% of AML patients. Increased PVT1 has been shown to induce necrosis and apoptosis in AML cell lines; this may be achieved via repressing the expression of the c-MYC gene [45]. This suggests that PVT1 overexpression in AML cells could set off biological reactions that suppress growth and initiate programmed cell death [46,47].

HOTAIRM1, a myeloid-specific lncRNA located between HOXA1 and HOXA2, regulates myeloid maturation by modulating integrin genes such as CD49d and CD11c. Its downregulation impairs ATRA-induced granulocyte differentiation [48], and its position within the HOXA cluster suggests a broader regulatory role over nearby HOXA genes [49].

Nevertheless, additional research is required to completely clarify the degree of HOTAIRM1′s regulatory impact on nearby genes. The complex role that HOTAIRM1 plays in myeloid differentiation pathways underscores its potential importance as a molecular regulator of AML. Table 2 lists lncRNAs recently reported in acute myeloid leukemia. Figure 2 demonstrates an illustration of the role of lncRNA in different cellular processes in AML initiation.

## 4. Role of circRNAs in AML

A class of ncRNAs known as circRNAs is distinguished by its circular structure. The circRNAs create closed circular RNA molecules because they lack separate 3′ and 5′ ends, in contrast to linear RNAs. circRNA and lncRNA, having common properties such as being non-coding and highly conserved and having a long life span, also share common functions like serving as a sponge for miRNA [97,98]. The altered expression of circRNAs has been associated with various stages of tumorigenesis, metastasis, and drug resistance [98].

Since it was initially noticed in viroids, this unusual circular arrangement is a characteristic shared by all eukaryotic cells. It is becoming increasingly evident that circRNAs regulate gene expression and impact many facets of leukemogenesis, including adhesion, proliferation, cell cycle changes, and apoptosis [99]. The expression of circRNA is also highly cell-type- and development-stage-specific, and circRNA directly interacts with proteins and modulates their function [100,101,102,103]. The avoidance of apoptosis seems to be a crucial element in leukemogenesis in the case of AML. A recent investigation brought the importance of circ-DLEU2 (hsa_circ_0000488) in AML to light [103]. When compared to healthy controls, the expression levels of this circRNA were greater in AML patients. Notably, there was a correlation found between enhanced cell proliferation in vitro, faster tumor growth in vivo, and decreased apoptosis when there was elevated circ-DLEU2 expression. By serving as a sponge for miRNA-496, circ-DLEU2 was responsible for increasing the expression of the PRKACB gene. It is noteworthy that PRKACB encodes the catalytic subunit of cyclic AMP-dependent protein kinase, a protein crucial in regulating various cell signaling processes.

In a recent study, the bone marrow of AML patients compared to the bone marrow of healthy controls demonstrated an increase in circ-vim expression using RT-qPCR. The authors of this study suggested the critical role of Circ-Vim as a tumor promoter based on their findings [104]. In another study, circ-Foxo3 was found to be downregulated in de novo AML patients. The authors also reported a positive correlation between circ-Foxo3 expression and *Foxo3* gene expression [105]. In a separate study, a pattern of expression of five circRNAs was reported, among which two were upregulated (circ_0033381 and circ_0049657), while three (circ_0001187, circ_0008078, and circ_0001947) were downregulated. This pattern distinguished high-risk and low-risk AML patient cohorts, indicating the potential of circRNA as a promising diagnostic biomarker [106]. Among AML patient cohorts compared to healthy controls with iron deficiency, the authors observed an upregulation in circ_009910 and circ-ANAPC7. In another study, circ-0009910 was reported as a critical regulator of cell cycle progression, proliferation, and apoptosis in leukemic cells by acting as an miR-Loa-5p [107,108].

In one study, has_circ_0012152 was observed with an increased expression level in AML patients compared with ALL patients [109]. In a separate study, 273 and 296 circRNAs showed upregulation and downregulation, respectively, in a cohort of pediatric AML patients compared with healthy controls [110]. Circ_004136, expressed from the RING protein 13 (RNF13) gene, showed very significant upregulation in pediatric AML patients. Another study also reported the upregulation of has_circ_0004136 (circRNF13) in AML patients [110]. The decreased expression level of circRNF13 was positively correlated with reduced proliferation, cell cycle arrest, increased apoptosis, and the attenuation of migration and invasion of MAL cells by serving as a sponge for miRNA1224-5p. In a separate study, the expression of has_circ_100290 (encoded by the parent gene SLC30A7) was observed to be increased in AML patients [111]. In the cell line model system, knockdown of has_circ100290 inhibited cell proliferation, promoted apoptosis, and targeted miR-203. This resulted in the regulation of cyclinD1, CDK4, BCL2, and cleaved Caspase-3 expression, regardless of the host gene SLC30A7 expression, which was unchanged in this knockdown model [111]. In another study, knockdown of has_circ_0002483 led to reduced cell proliferation, cell cycle arrest (G0/G1 phase), and elevated apoptosis by decreasing BCL2 and increasing BAX and C-caspase-3 [112]. Similarly, the reduced expression level of has_circ_007980-KD was associated with lower viability and increased apoptosis in AML cells [113]. Another circPAN3 showed increased expression level in relapsed/refractory AML compared to chemosensitive AML patients, though the parent gene expression did not follow a similar pattern [114]. Likewise, circ-ANXA2 expression was observed to be increased in AML cases [115], which was correlated with a poor-risk phenotype in AML patients. In vitro, knockdown of circ-ANXA2 was associated with enhanced apoptosis, reduced proliferation of MAL cells, and increased chemosensitivity to cytarabine and daunorubicin [115]. AML patients showed overexpression of both circ_0000370 and its parent gene FLI-1, according to a study by Zhang et al. [116].

Additionally, circ_0000370 expression was shown to be higher among AML patients with FLT3-ITD mutations than in FLT3 wild-type (WT) patients, and this elevated expression was linked to a worse prognosis. The study additionally demonstrated a positive association between circ_0000370 expression and FLT3-ITD expression. Furthermore, the downregulation of circ_0000370 expression was seen following the injection of quizartinib, an inhibitor of FLT3 [116]. This implies that FLT3-ITD and circ_0000370 may have a regulatory interaction in which the inhibitor affects the expression of the circRNA. circ_0000370 was discovered to control both apoptosis and cell survival functionally. This is accomplished by its ability to function as a sponge for miR-1299, which raises S100A7A expression. S100A7A is known to be involved in several cellular functions. Its overexpression, which is mediated by circ_0000370, raises the possibility that this circRNA plays a role in the deregulation of survival of cells and apoptosis in AML [116]. In summary, circRNAs demonstrate potential in AML as both diagnostic and prognostic biomarkers, as well as for monitoring disease status and therapeutic response. Table 3 summarizes a list of circRNAs having a regulatory role in AML.

## 5. Role of miRNAs in AML

Extensive profiling and functional studies have shed light on the sensitive regulatory role performed by miRNAs in hematopoiesis. In the past decade, the role of miRNAs in AML has been studied very exhaustively. However, there is less literature available on the other two ncRNAs (lncRNA and circRNA). That is why the primary focus of this review is to investigate lncRNA and circRNA. At the same time, we would like to discuss the role of miRNAs in the pathogenesis of AML. miRNAs play a critical regulatory role through various means, including epigenetic alterations, copy number alterations, aberrant targeting of the promoter region, and alterations in the vicinity of oncogenic genomic regions due to chromosomal rearrangements. It has been recently observed that lncRNAs and circRNAs in AML can modulate the function of specific miRNAs and promote the initiation, maintenance, and progression of leukemogenesis [120]. Consequently, it is not surprising to find that a wide range of microRNAs exhibit different expression patterns when comparing the blasts of AML to their counterparts in healthy cells. These miRNAs play a critical role in the pathophysiology of AML. They do this by acting through a variety of mechanisms that together affect the onset and course of this hematological cancer [121]. Variations in copy number are the primary way that microRNAs contribute to the pathogenesis of AML. Variations in the quantity of distinct microRNAs might result in disequilibrium within regulatory networks, hence upsetting regular hematopoietic functions and encouraging the development of AML [122]. Second, chromosomal translocations can change the distance that microRNAs have from cancer-causing genomic regions. These translocations can move these regulatory molecules close to genes that may promote malignancy, which might result in the aberrant regulation of gene expression and worsen the development of AML [123]. Other mechanisms through which miRNAs play a role in AML include epigenetic modifications, abnormal targeting of miRNA promoter regions by modified transcription factors or oncoproteins, and the disruption of miRNA processing [120,124].

A prevalent issue in AML known as chemoresistance has been connected to miRNAs. These miRNAs influence apoptosis, the cell cycle, and the function of specific proteins, such as ATP-binding cassette transporters, that aid cells in discharging chemotherapy medicines, among other mechanisms that contribute to resistance. Li and colleagues found that, in comparison to normal leukemia cells (K562 cells), miR-181a was less active in a subset of leukemia cells (K562/A02 cells) that were resistant to doxorubicin [125]. By targeting particular genes (BCL-2 and MCL-1), they increased the activity of miR-181a in the resistant cells, which helped increase the cells’ susceptibility to doxorubicin. By lowering their levels through the use of miR-181a, the cells showed an improved response to the treatment. Similarly, it was discovered that miR-181a was downregulated in another kind of leukemia cell line (HL-60/Ara-C) that was resistant to the medication Ara-C. Through the promotion of apoptosis, the upregulation of miR-181a in these cells enhanced their reactivity to Ara-C treatment. This was accomplished by releasing cytochrome c and turning on several proteins that cause cell death, including caspase-9 and caspase-3. Once more, BCL-2 was the target of miR-181a, which kept these resistant cells from dying [126].

AML is diagnosed chiefly using the widely accepted molecular and cytogenetic criteria published by the WHO in 2016. Notably, the complex association these criteria uncover between AML subtypes and unique miRNA signatures correlated with each further emphasizes the importance of these criteria. There is intrinsic variation in AML when it comes to miRNA expression profiles. Among various AML subtypes, miRNAs frequently have distinct signatures that differentiate them from one another [120]. These unique miRNA signatures have the potential to be highly effective prognostic and diagnostic markers, assisting in the differentiation of AML subtypes, forecasting disease prognoses, and directing therapeutic approaches. Table 4 highlights various miRNAs in association with AML subtypes and their role in the pathogenesis of AML.

## 6. Interplay and Crosstalk Among ncRNA Classes in AML Pathogenesis

The complex pathogenesis of AML is increasingly recognized to involve intricate regulatory networks governed by diverse classes of ncRNAs, including lncRNAs, miRNAs, and circRNAs. Rather than acting in isolation, these ncRNAs frequently engage in dynamic crosstalk, forming competing ceRNA networks that modulate gene expression and cellular behaviors relevant to leukemogenesis [144]. The competing ceRNA hypothesis was put forth by Salmena et al. In 2011, it was indicated that lncRNAs bind to endogenous miRNAs in AML in a competitive manner [120].

### 6.1. lncRNA-miRNA Interaction

Recent research has shown that the aberrant expression of circular and lncRNAs in AML may alter the roles of certain miRNAs, aiding in the start, upkeep, and progression and development of leukemogenesis. Several lncRNAs function like molecular sponges for miRNAs, sequestering them away from their target mRNAs [97,145]. This sponging mechanism can derepress certain oncogenes or mute tumor suppressors based on the targets. For example, the lncRNA XIST sponges miR-29a, thus enhancing MYC expression and inducing AML cell proliferation [30]. UCA1 sequesters miR-125a, contributing to drug resistance in AML cells. Specific lncRNAs also influence miRNA processing or stability [146]. For instance, nuclear lncRNAs may bind to Drosha or Dicer complexes, modulating miRNA maturation. Dysregulation of this process may shift the balance of tumor-suppressive vs. oncogenic miRNAs in AML.

### 6.2. CircRNA-miRNa-mRNA Axis

Emerging evidence suggests that lncRNAs and circRNAs can regulate each other’s expression or stability through shared miRNA response elements (MREs) [147,148]. For example, both may sponge the same miRNA, thereby cooperatively regulating downstream mRNA targets. In this way, ncRNA species can amplify or buffer oncogenic signals in AML. circRNAs were reported to function as miRNA sponges in complex endogenous RNA networks [120], thus influencing AML progression. For example, circ_0009910 is reported to bind mir-491-5p, leading to the upregulation of MAPK1, which enhances AML cell viability and proliferation. Such interactions underscore the functional convergence between circRNAs and incRNAs in miRNA regulation. The lncRNA H19, which was found to be significantly elevated in bone marrow biopsies from individuals suffering from AML-M2, serves as an illustration of this interaction [64]. As a competitive endogenous RNA, H19 acts to sequester miR-19a/b and stimulate the growth of AML cells. CircRNAs have been the subject of several studies recently due to their function as “miRNA sponges” in intricate endogenous RNA networks. As an example, circRNA HIPK2 has been found to function as a miR-124-3p sponge, controlling the differentiation of NB4 cells triggered by all-trans retinoic acid (ATRA) [118]. CircANAPC7 was shown to be considerably elevated in AML in the research conducted by Chen et al. [119]. Through the utilization of an Arraystar human circRNAs microarray and bioinformatics studies, it was anticipated that ANAPC7 would bind to the miR-181 family, suggesting a possible role for it in the pathophysiology of AML. This implies that circANAPC7 may operate as a miRNA sponge, adjusting miRNA activity and affecting important pathways involved in the genesis of AML. These results demonstrate the complex regulatory involvement of lncRNAs and circRNAs in AML by functioning as ceRNAs or miRNA sponges and influencing the production and activity of particular miRNAs linked to the pathophysiology of leukemia [149]. However, due to the vast role of non-coding RNAs in cellular processes, it is tough to study the connection network of the three ncRNAs, which is why very little literature is available on this area. A future study is required to explore the complex regulatory role of the three ncRNAs. Gaining knowledge of these intricate RNA networks can help identify possible therapeutic targets for the treatment of AML.

## 7. LncRNAs: Emerging Clinical Utility

lncRNAs are gaining attention for their important ability to regulate gene expression via several means, like chromatin remodeling, ceRNA activity, and scaffold functions [150]. Studies established that lncRNAs play a critical role in AML, and important lncRNAs were HOTAIRM1, MALAT1, NEAT1, MEG3, PVT1, and HOTAIR. Apart from this, MALAT1 and NEAT1 have been shown to regulate the apoptosis, proliferation, and differentiation of AML blasts [151]. HOTAIRM1 interacts with the HOXA gene cluster and is very specific to myeloid cells, making it an important diagnostic biomarker [49]. Several lncRNAs (e.g., UCA1 and MIR100HG) have been reported in chemoresistance, especially to cytarabine and anthracyclines [29,91,92,93]. Unlike miRNAs, lncRNAs often express tissue- and subtype-specific expression, increasing their importance as precision biomarkers [152,153]. Understanding the crosstalk among different classes of ncRNAs in AML has strong potential for diagnostic biomarker identification and the development of ncRNA-targeted therapies. Disrupting specific classes of lncRNA–miRNA–mRNA interactions may offer novel therapeutic targets to overcome drug resistance or halt leukemic progression.

The role of miRNAs has been extensively investigated in AML for their role in leukemogenesis, prognosis, and therapy resistance [154,155]. Several miRNAs are now established as diagnostic and prognostic biomarkers; for example, miR-155, miR-181a, and the miR-29 family are among the most frequently dysregulated miRNAs in AML pathogenesis [124,156]. MiR-29a expression downregulates DNMT3A and promotes differentiation, and studies showed that it is associated with better prognosis [157,158]. The miR-155 is overexpressed in the high-risk AML category and is linked with poor survival and therapy resistance [159,160]. Several validated targets of miRNAs in AML pathogenesis include FLT3, NPM1, CEBPA, MLL, and DNMT3A [124]. Therapeutically, antagomiRs and miRNA mimics are currently in the preclinical stage to reverse pathogenic miRNA effects, although clinical application still remains limited. Though relatively underexplored compared to miRNAs and lncRNAs, circRNAs are increasingly recognized for their miRNA sponge activity, regulatory stability, and role in drug resistance: circ_0009910, circRNF220, and circMYBL2 are differentially expressed in AML. circ_0009910 has been linked to FLT3-ITD+ AML, acting through sponging miR-20a-5p. circRNF220 modulates AML progression via the miR-30a–SLC7A5 axis, suggesting therapeutic implications [161,162]. However, most circRNAs in AML still lack functional validation, and clinical translation is pending.

## 8. Conclusions and Future Directions

ncRNAs play an important role in development, progression, and drug resistance in hematological malignancies. However, due to the limited number of studies relating to all three ncRNA networks, their application in clinical practice is still at its preliminary stages. The crosstalk between different ncRNA classes—especially lncRNAs, miRNAs, and circRNAs—plays an important role in orchestrating the gene expression pattern underlying AML. These interactions among different class of ncRNAs showed an additional and very important layer of post-transcriptional regulation that contributes to leukemic progression and therapy response. Future research is required that will systematically delineate these interactions diligently and explore them for potential therapeutic targeting in AML. While miRNAs remain the most clinically studied class of ncRNAs in AML, lncRNAs are rapidly gaining traction, especially in terms of prognostic potential and therapy resistance mechanisms. circRNAs also offer exciting possibilities due to their stability and specificity, but functional validation is still at the early stage. Future studies that will integrate multi-omics, single-cell sequencing, and ncRNA-targeted therapeutics are essential to explore the full therapeutic potential of these RNA species in AML. With miRNAs, this approach is the most sophisticated and may be applied in two different ways [163]. Firstly, ASOs can be used to silence oncomiRNAs, which are overexpressed oncogenic miRNAs. Chemical alteration, such as the usage of locked nucleic acids, is frequently used to accomplish this. On the other hand, the second strategy aims to bring inhibited tumor-suppressor miRNAs back to life [164]. Since most miRNAs affect a wide range of downstream targets, the regulation of specific target ncRNAs is particularly intriguing. This has the potential to reduce the danger of acquired resistance while also improving treatment effectiveness in theory [165]. At the same time, direct ncRNA modulators are not yet the subject of registered clinical studies for AML. Advancements in miRNA-directed therapeutics for different kinds of cancer point to significant advancements in AML in the coming years. The foundation established by developments in other cancer entities underscores the tremendous potential for future treatments in AML, even in the absence of ongoing research in direct ncRNA modification for AML therapy. It is essential to perform functional and molecular characterization of lncRNA before using it as a therapeutic tool. This will ensure the desired efficacy and safety profiles of lncRNA. Novel therapeutic approaches using the direct involvement of ncRNAs are expected to rise as our comprehension of the roles of ncRNAs in leukemia deepens, unlocking new possibilities for the development of efficient and tailored AML therapeutics. Numerous studies suggest that ncRNAs are promising diagnostic tools for AML. In cytogenetically normal AML patients, upregulation of let-7a-2-3p and downregulation of miR-188-5a have been reported. These findings not only suggest a possible diagnostic marker but also demonstrate correlations with extended overall survival (OS) and event-free survival (EFS) [166]. Moreover, miR-181 expression levels in AML have shown promise for diagnosis. In AML, downregulation of toll-like receptor, interleukin 1β, HOXA7, and PBX3 is associated with a high expression of miR-181 [131].

Additionally, medication resistance has been linked to miR-181b downregulation in AML patients [167]. According to this, assessing the expression levels of particular miRNAs may offer insightful diagnostic information about AML and open the door to the development of more precise and successful diagnostic techniques in clinical settings. Since lncRNAs are exclusive to specific tissues and disorders, they are becoming more and more recognized as viable biomarkers for prognostic and diagnostic applications. Because of their varying expression levels in normal tissue, lncRNAs are useful markers or predictors of disease stage [168,169]. Researchers used RNA sequencing to measure the expression of long non-coding RNA (lncRNA) in 274 AML patients receiving rigorous treatment from a Swedish cohort in a thorough investigation [170]. Determining the presence of lncRNA subtypes and their predictive value was the goal. The study used an independent patient cohort to validate its classification of lncRNAs into four categories. The findings demonstrated the potential of lncRNA in providing crucial data for AML patients.

In conclusion, ncRNAs regulate complex regulatory networks that play a key regulatory role in the pathophysiology of acute myeloid leukemia. Several ncRNA expressions have been reported with diagnostic and prognostic potential, supporting clinicians in subtype categorization, prognosis evaluation, and drug therapy response prediction. Despite a lot of research, the mechanisms behind the regulatory role of miRNAs in AML pathogenesis are still intricate and multi-dimensional due to the wide spectrum of gene expression patterns. LncRNAs and cirRNAs have gained more attention in past years. Targeting ncRNAs for therapeutic targets in AML appears increasingly intriguing as our current understanding of these multi-dynamic interactions grows, opening the door to more efficient and focused treatment approaches down the road.

Recent research has made it clear that the abnormal expression of circular and lncRNAs in AML may alter the roles of particular miRNAs, aiding in the start, upkeep, and progression of leukemogenesis. The competing endogenous RNA (ceRNA) hypothesis was put forth by Salmena et al. in 2011, indicating that lncRNAs bind to endogenous miRNAs in AML in a competitive manner [120]. The circRNAs were reported to function as miRNA sponges in the complex endogenous RNA networks [120]. The lncRNA H19, which was found to be significantly elevated in bone marrow biopsies from individuals suffering from AML-M2, serves as an illustration of this interaction [64]. As a competitive endogenous RNA, H19 acts to sequester miR-19a/b and stimulate the growth of AML cells. CircRNAs have been the subject of several studies recently due to their function as “miRNA sponges” in intricate endogenous RNA networks. As an example, circRNA HIPK2 has been found to function as an miR-124-3p sponge, controlling the differentiation of NB4 cells triggered by all-trans retinoic acid (ATRA) [118]. CircANAPC7 was shown to be considerably elevated in AML in the research conducted by Chen et al. [119]. Through the utilization of an Arraystar human circRNA microarray and bioinformatics studies, it was anticipated that ANAPC7 would bind to the miR-181 family, suggesting a possible role for it in the pathophysiology of AML. This implies that circANAPC7 may operate as an miRNA sponge, adjusting miRNA activity and affecting important pathways involved in the genesis of AML. These results demonstrate the complex regulatory involvement of lncRNAs and circRNAs in AML by functioning as ceRNAs or miRNA sponges and influencing the production and activity of particular miRNAs linked to the pathophysiology of leukemia. However, due to the vast role of non-coding RNAs in cellular processes, it is tough to study the connection network of the three ncRNAs, which is why very little literature is available in this area. A future study is required to explore the complex regulatory role of the three ncRNAs. Gaining knowledge of these intricate RNA networks can help identify possible therapeutic targets for the treatment of AML.

## 9. Concluding Remarks

ncRNAs play a critical role in development, progression, and drug resistance in hematological malignancies. However, due to the limited number of studies relating to all three ncRNA networks, their application in clinical practice is still at its early stages.

Future developments for ncRNAs in AML therapy are promising, with a focus on modulating their expression as specific targets for therapeutic interventions. Novel therapeutic approaches involving the direct manipulation of ncRNAs are expected to emerge as our comprehension of the roles of ncRNAs in leukemia deepens, opening up new possibilities for the development of efficient and tailored AML therapeutics.

## Figures and Tables

**Figure 1 ncrna-11-00070-f001:**
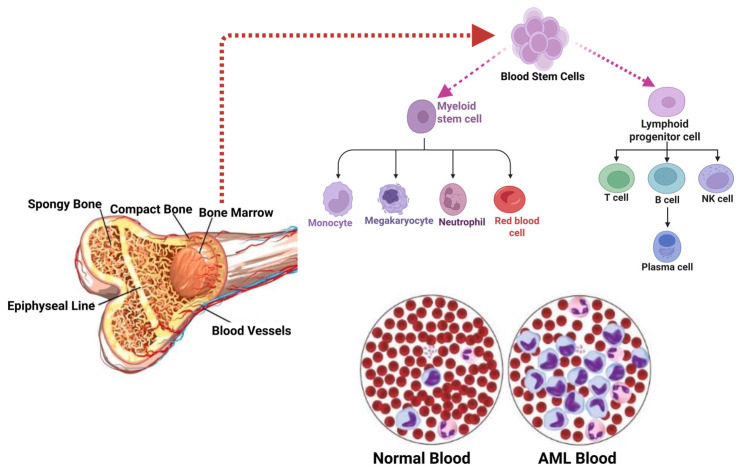
AML begins in the bone marrow. The spongy tissue found inside bones is called bone marrow, and it is here that blood cells are produced. Hematopoietic stem cells (HSCs) are found in the red marrow, where they undergo distinct progenitor stages before differentiating into red blood cells, white blood cells, and platelets. AML begins when a single, immature white blood cell known as a “blast” has several alterations that enable unchecked cell division.

**Figure 2 ncrna-11-00070-f002:**
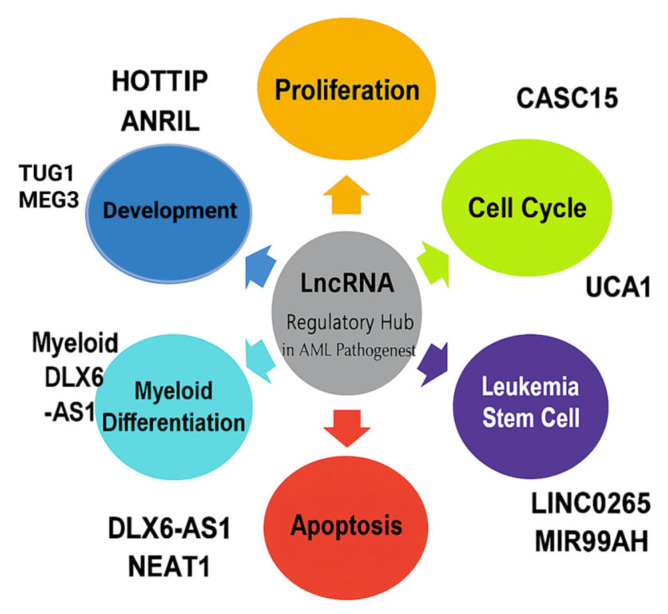
LncRNA-mediated diverse regulatory function in AML. This schematic diagram illustrates the diverse functioning of lncRNAs and their influence on regulation. The dysregulation of these ncRNAs can affect oncogene or tumor suppressor gene expression, ultimately contributing to leukemogenesis and cancer progression (lncRNAs mentioned here discussed in the main text).

**Table 1 ncrna-11-00070-t001:** WHO classification of acute myeloid leukemia.

Group Number	AML Type	Major Clinical Characteristics	Commonly Implicated Genes
1	AML with recurrent genetic abnormalities	Distinct genetic alterations.	FLT3, NPM1, CEBPA, RUNX1, etc.
		Specific implications for treatment.	
		Varied prognosis.	
2	AML with myelodysplasia-related changes	Overlapping features with myelodysplastic syndrome (MDS).	TP53, ASXL1, EZH2, etc.
		Impacts on therapy choices.	
3	Therapy-related myeloid neoplasms	Arises as a result of prior cancer treatments.	MLL rearrangements, TP53, etc.
		Special considerations in diagnosis and therapy.	
4	AML not otherwise specified	Broad category encompassing AML cases without specific features.	Various gene mutations
		Diagnosis and treatment are more generalized.	
		Variable clinical presentations.	
5	Myeloid sarcoma	Manifests as extramedullary tumors in various body tissues.	No single common gene implicated
		Diverse symptoms depending on the location of the tumors.	
		Clinical complexity due to tumor heterogeneity.	
6	Myeloid proliferations related to Down syndrome	Associated with individuals with Down syndrome.	GATA1 mutations
		Diagnosis and treatment tailored to this patient group.	

**Table 2 ncrna-11-00070-t002:** List of lncRNAs identified in acute myeloid leukemia.

Name	Key Observation	Reference
ANRIL	Oncogenic in nature, upregulated in AML, promotes proliferation, targets adipoR1 and miT-34/HFAC1, and maintains glucose metabolism	[50,51]
CASC15	Oncogenic in nature, upregulated in RUNX1 rearranged AML, and activates SOX4 via regulating YY1 transcription factor	[52]
CCAT1	Oncogenic in nature, upregulated in AML, and suppresses monocytic differentiation and promotes proliferation	[53,54]
CCDC26	Upregulated, tumor suppressor, age-associated expression, reduces c-Kit expression, regulates AML cell proliferation, and induces drug resistance	[55,56,57]
CDK6-AS1	Oncogenic in nature, it reduces RUNX1 transcription and is associated with a higher expression level, which is associated with poor treatment response	[58]
CRNDE	Upregulated in AML and regulates NOTCH2 in acute promyelocytic cells	[59]
DARS-AS1	Oncogenic in nature and higher expression is associated with poor survival	[60]
DLEU2	Tumor suppressor in AML	[61]
DUBR	Induces proliferation in AML cells	[62]
GAS6-AS2	Oncogenic in nature and regulates GAS1 and AXL expression	[63]
H19	Oncogenic in nature	[64,65,66]
HOTAIR	Upregulated, oncogenic in nature, and promotes proliferation and differentiation in AML cells	[26,67,68]
HOTAIRM1	Upregulated in AML cells, regulates myeloid differentiation, and is a tumor suppressor	[31,49,67]
HOXA10-AS	Upregulated, HSC-specific lncRNA that induces proliferation	[27]
HOX-AS2	Oncogenic in nature and apoptosis repressor	[69,70]
HOXB-AS3	Upregulated and regulates proliferation in leukemic cells	[71]
HOXBLINC	Upregulated as a chromatin modulator	[72]
IRAIN	Downregulated in Mal cells and restricts tumor cell migration	[73,74]
LAMP5-AS1	Upregulated and maintains methyl transferase activity	[75]
LINC00152	Regulate PARP1	[32]
LINC00641	Oncogenic in nature	[76]
LINC00998	Tumor suppressor and reduced expression associated with poor survival	[77]
LINC01257	Oncogenic in nature and higher expression associated with poor survival	[78]
LINC-223	Oncogenic in nature and induces differentiation	[79]
Lnc-SOX6-1	Oncogenic in nature, and increased expression is associated with poor survival	[80]
LONA	Upregulated, promotes leukemogenesis, and is involved in differentiation	[25]
LOUP	Downregulated and tumor Suppressor	[35]
MAGI2-AS3	Tumor suppressor	[36]
MEG3	Tumor suppressor	[81]
MIR100HG	Oncogenic in nature	[43]
MONC	Oncogenic	[43]
MORRID	Upregulated in AML cells and induces proliferation	[82]
MVIH	Oncogenic, and increased expression affects treatment response	[83]
NEAT1	Tumor suppressor regulates myeloid differentiation	[37]
NR-104098	Tumor suppressor	[34]
PANDAR	Upregulated in AML	[38]
PU.1-AS	Oncogenic	[84]
PVT-1	Oncogenic	[44,46,54]
RUNXOR	Upregulated in AML and oncogenic	[85]
SNHG5	Upregulated in AML	[86]
SNHG14	Oncogenic in nature	[87]
TUG1	Upregulated in AML cells, oncogenic, and induces proliferation	[88,89,90]
UCA1	Upregulated in CN-AML and maintains leukemic cell proliferation, migration, and invasion	[29,91,92,93]
LINC00909	Oncogenic and higher expression associated with poor survival	[94]
WT1-AS	Tumor suppressor	[95]
XIST	DOX resistance	[30]
XLOC_109948	Higher expression is associated with poor prognosis	[96]

**Table 3 ncrna-11-00070-t003:** Role of circRNAs in AML.

circRNAs	Role in AML	Reference
f-circPR	Promotion of proliferation and colony formation in leukemia cells	[117]
circ_100290	Upregulation of proliferation and inhibition of apoptosis utilizing the miR-203/Rab10 axis	[118]
circ_0009910	Upregulation of proliferation, inhibition of apoptosis, and prediction of poor prognosis	[107]
circ_DLEU2	Promotion of AML cell proliferation and inhibition of cell apoptosis and AML tumor formation in vivo via suppression of miR-496 and promotion of PRKACB expression	[61,103]
circ_HIPK2	Influence on ATRA-induced differentiation of APL cells and impairment of AML1- and p53-mediated transcription	[118]
has_circ_0004277	Elevation of hsa_circ_0004277 is associated with chemotherapy	[106]
f-circM9	Promotion of proliferation and colony formation in leukemia cells	[117]
circ_0000370	Upregulated in AML, i.e., correlated with poor prognosis	[116]
circ_vim	Tumor promoter	[104]
circ_Foxo3	Downregulated in AML cells	[105]
circ_009910	Upregulated in AML cells and a critical regulator of cell cycle progression, proliferation, and apoptosis in leukemic cells.	[107,108]
circ_ANAPC7	Upregulated in AML cells	[119]
has_circ_0012152	Upregulated in AML cells	[109]
Circ_004136	Upregulated in pediatric AML	[110]
has_circ_100290	Upregulated in AML patients	[111]
has_circ_002483	Upregulated in AML patients	[112]
has_circ_007980	Reduces the expression associated with the lower viability of leukemic cells	[113]
circPAN3	Upregulated in AML cases	[118]

**Table 4 ncrna-11-00070-t004:** Role of miRNAs in subtypes of AML.

miRNA	AML Subtype	Role in AML	References
miR-9	MLL-rearranged AML	Overexpressed, it promotes cell growth and inhibits apoptosis	[127]
miR-9	Pediatric AML with t(8;21)	Downregulated; acts as a tumor suppressor and induces differentiation	[128]
miR-9-1	t(8;21) AML	Downregulation and overexpression induce differentiation and inhibit proliferation	[129]
miR-10b	NPM1 and DNMT3A mutation AML	Correlated with higher BM blast percentage	[130]
miR-181	CN-AML with CEBPA mutations, FLT3-ITD, t(15;17)	Overexpressed	[131]
miR-155	FLT3-ITD-associated AML	Upregulated; targets PU.1; knockdown represses proliferation and induces apoptosis	[132]
miR-122	FAB subtype M7	Downregulation and overexpression inhibit cell proliferation	[133]
miR-196b	Pediatric AML with M4/5 subtypes	Higher expression predicts a poor outcome	[134]
miR-195	FAB-M7, unfavorable karyotypes	A decrease in BM and serum is associated with these factors	[135]
miR-135a		Downregulated in AML	[136]
miR-144-3p		Antiapopotic	[137]
miR-150		Suppressed hematopoiesis via releasing exosomes loaded with miRNA	[138]
miR-193a		Induces the oncogenic activity of AML-ETO	[139]
miR-193b		Induces apoptosis and block G1/S phase	[140]
miR-339-5p		Inhibits the cell proliferation of AML cells	[141]
miR-370		Activates the RAS signaling pathway	[142]
miR-375		Playing a role in DNA hypomethylation	[143]

## Data Availability

Not applicable.

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
