# Peer review of "Role of Non-Coding RNAs in Acute Myeloid Leukemia"

_ncrna, 2025, doi:10.3390/ncrna11050070_

Round 1
Reviewer 1 Report
Comments and Suggestions for Authors
The roles of ncRNAs in AML is an important topic for the field of AML research. It is clear that there are numerous ncRNA species and specific transcripts which play important roles in the pathology of AML. Unfortunately, this manuscript suffers from organizational and grammatical shortcomings that detract from the message of the authors.
Line 73: lncRNAs12. instead of long noncoding RNAs.12
Line 76: ncRNAs (already defined the acronym on line 69)
Line 79-80: complementary target RNAs. miRNAs can bind other RNA than just mRNA. lncRNAs frequently act as sponges for miRNAs.
Line 82: assembly14. not assembly14.
Line 89: lncRNAs acronym already defined.
Line 96-97: Sentence unclear and needs rephrasing.
Line 102-104: Citation for LONA is needed.
Line 133: NEAT1 needs citation.
Line 136-137: Improve clarity.
Line 171-175: lacks citations and clarity/justification.
Figure 2: Oversimplification and lacks detail. Does not add to the understanding or interpretation of the roles of lncRNAs in AML.
Figure 3: Again, an oversimplification of the roles of ncRNAs in AML. Seems to implicate circRNAs and lncRNAs are globally down regulated in AML. Significant revision or removal would be appropriate.
Not all corrections are marked due to the volume of issues. The above suggestions are examples and not exhaustive. There are frequent typographical errors and organizational issues with unnecessary repeats. General copy editing is needed to improve clarity, brevity, and accuracy. Use of acronyms is inconsistent. There are also frequent shifts in tense which affect clarity and readability (see line 114-115). There are grammatical errors and inconsistencies which need to be corrected. There are also frequent statements or claims made without sufficient evidence or reasoning provided.
Additionally, there is a general lack of completed thoughts and incomplete thoughts or a lack of explanation for the significance or implications of stated findings. The authors need to work on the overall organization to make sure findings stated flow logically and are appropriate to the point they are trying to make.
Check that references formatting conform to the required format of the journal. Reference formatting is inconsistent.
Comments on the Quality of English LanguageThere are significant organizational issues with this manuscript which need to be addressed to ensure clarity and accurate communication of findings. Consistency of grammar, organization, and tone/tense needs to be improved to ensure accuracy and fidelity in communicating the authors' message.
Author Response
Reviewer 1:
The roles of ncRNAs in AML is an important topic for the field of AML research. It is clear that there are numerous ncRNA species and specific transcripts which play important roles in the pathology of AML. Unfortunately, this manuscript suffers from organizational and grammatical shortcomings that detract from the message of the authors.
Comment 1:
Line 73: lncRNAs12. instead of long noncoding RNAs.12
Line 76: ncRNAs (already defined the acronym on line 69)
Line 79-80: complementary target RNAs. miRNAs can bind other RNA than just mRNA. lncRNAs frequently act as sponges for miRNAs.
Line 82: assembly14. not assembly14.
Line 89: lncRNAs acronym already defined.
Line 96-97: Sentence unclear and needs rephrasing.
Line 102-104: Citation for LONA is needed.
Line 133: NEAT1 needs citation.
Line 136-137: Improve clarity.
Line 171-175: lacks citations and clarity/justification.
Response: we deeply appreciate for the respected reviewer’s critically reading of the manuscript and his/her valuable time and thankfull for the comments, As respected reviewer advised, all the suggested modifications have been incorportaed, and highlighted in revised version of MS.
Comment 2: Figure 2: Oversimplification and lacks detail. Does not add to the understanding or interpretation of the roles of lncRNAs in AML.
Response: We appreciate the reviewer for this valuable feedback. Figure 2 has been substantially revised to provide a more comprehensive and detailed representation of the roles of lncRNAs in AML. Corresponding figure legend and text have been expanded to enhance clarity and interpretation.
Comment 3 : Figure 3: Again, an oversimplification of the roles of ncRNAs in AML. Seems to implicate circRNAs and lncRNAs are globally down regulated in AML. Significant revision or removal would be appropriate.
Response: We appreciate the reviewer’s insightful comment. As suggested we align with respected reviewer’s suggestion that removal of this figure is appropriate and this removal further enhance the quality of manuscript.
Comment 4: Not all corrections are marked due to the volume of issues. The above suggestions are examples and not exhaustive. There are frequent typographical errors and organizational issues with unnecessary repeats. General copy editing is needed to improve clarity, brevity, and accuracy. Use of acronyms is inconsistent. There are also frequent shifts in tense which affect clarity and readability (see line 114-115). There are grammatical errors and inconsistencies which need to be corrected. There are also frequent statements or claims made without sufficient evidence or reasoning provided.
Additionally, there is a general lack of completed thoughts and incomplete thoughts or a lack of explanation for the significance or implications of stated findings. The authors need to work on the overall organization to make sure findings stated flow logically and are appropriate to the point they are trying to make.
Check that references formatting conform to the required format of the journal. Reference formatting is inconsistent.
Response: We sincerely thankfull for the respected reviewer for highlighting all these significant concerns that really helps in to raise the standard of the review for the readers. The entire manuscript has undergone for thoroughly revision by all the authors to address typographical errors, grammatical inconsistencies, tense shifts, and unnecessary repetitions. We have carefully/diliginetly streamlined the organization of sections to ensure a logical flow of findings and improved clarity of arguments. Incomplete or ambiguous statements have been expanded/removed to provide full meaning, with emphasis on the significance and implications of reported findings. The use of acronyms has been standardized throughout the text for consistency. All references have been re-checked and reformatted to strictly conform to the journal’s reference style. We believe these changes substantially enhance clarity, brevity, and overall accuracy of the manuscript.

Reviewer 2 Report
Comments and Suggestions for Authors
Dear Dr. Maurya,
Thank you for the opportunity to review your manuscript, "Role of Noncoding RNAs in Acute Myeloid Leukemia." I have now had the chance to assess the work. While the topic is of significant interest, my review has identified several shortcomings related to the manuscript's structure and clarity that detract from its overall effectiveness.
The manuscript's structure is undermined by the questionable placement of Section 7, "Current Methods for Identification of ncRNAs." Its location late in the document interrupts the logical flow from future applications to the conclusion.
Furthermore, the lack of internal structure in Section 2, "Role of Long Non-Coding RNAs (lncRNAs) & Cancer," is problematic. This section is overly long and combines numerous distinct topics without clear divisions, which negatively impacts its readability and the reader's ability to follow the arguments presented.
There is also a general lack of integration in the discussion of the different ncRNA classes. Each is largely presented in isolation, leading to a fragmented narrative that fails to convey the interconnectedness of these regulatory networks until much later in the manuscript.
Finally, the clarity of the figures is a concern. The diagram in Figure 3, intended to illustrate the crucial lncRNA mechanism, is ambiguous and does not effectively communicate the complex biological interactions at play.
Author Response
Dear Dr. Maurya,
Thank you for the opportunity to review your manuscript, "Role of Noncoding RNAs in Acute Myeloid Leukemia." I have now had the chance to assess the work. While the topic is of significant interest, my review has identified several shortcomings related to the manuscript's structure and clarity that detract from its overall effectiveness.
Response: We appreciate the reviewer’s valuable feedback regarding the shortcoming of our manuscript. We have undertaken a thorough revision of the manuscript to address all suggested shortcoming to improve and enhance overall quality of manuscript.
Comment 1: The manuscript's structure is undermined by the questionable placement of Section 7, "Current Methods for Identification of ncRNAs." Its location late in the document interrupts the logical flow from future applications to the conclusion.
Response: We appreciate the reviewer’s observation regarding the position of Section 7: Current Methods for Identification of ncRNAs. This section has been relocated earlier in the manuscript. We believe this reorganization improves logical flow and ensures a smoother transition to the conclusion.
Comment 2: Furthermore, the lack of internal structure in Section 2, "Role of Long Non-Coding RNAs (lncRNAs) & Cancer," is problematic. This section is overly long and combines numerous distinct topics without clear divisions, which negatively impacts its readability and the reader's ability to follow the arguments presented.
Response: Thank you for your constructive comment. As suggested, Section 2 has been reorganized into clearly defined subsections, each focusing on a specific aspect of lncRNAs in cancer.
Comment 3: There is also a general lack of integration in the discussion of the different ncRNA classes. Each is largely presented in isolation, leading to a fragmented narrative that fails to convey the interconnectedness of these regulatory networks until much later in the manuscript.
Response: We thank the reviewer for pointing out these important issues. We agree with the reviewer that the earlier version presented miRNAs, lncRNAs, and circRNAs largely in isolation. To address this, we have completely revised the discussion/conclusion & future direction with more added information to emphasize the interconnectedness of these regulatory networks.
Comment 4: Finally, the clarity of the figures is a concern. The diagram in Figure 3, intended to illustrate the crucial lncRNA mechanism, is ambiguous and does not effectively communicate the complex biological interactions at play.
Response: We appreciate the reviewer’s valuable feedback, as other respected reviewer also suggest that removal of this figure is appropriate and this removal further enhance the quality of manuscript. We respect the reviewer’s valuable suggestion. For better clarity Figure 3 has been removed now, the removal further enhnaces the better theme comunication of the manuscript.

Reviewer 3 Report
Comments and Suggestions for Authors
This manuscript provides a comprehensive review of the current literature on the roles of noncoding RNAs (ncRNAs, including miRNAs, lncRNAs, and circRNAs) in the complex landscape of acute myeloid leukemia (AML). The review covers an important and timely topic on cancer biology with a focus on ncRNAs, which is fitted well with the scopes of the journal. However, the current version of the manuscript requires significant revision to enhance its analytical depth, clinical relevance, and clarity. Addressing the points above would greatly improve the manuscript’s quality and impact.
Major concerns
- The manuscript lacks critical discussion in some sections. While many ncRNAs are listed, the review does not sufficiently analyze or compare their relative strengths or clinical utility. A more analytical, less descriptive approach is recommended.
- The section of “Mechanisms of ncRNAs in AML” needs to be expanded with better depth on the analysis of how ncRNAs involved in the more biological insight in the development of AML. More mechanistic analysis is required.
- The manuscript should more clearly address how lncRNAs are being translated into clinical applications (add a new section). Are there any ongoing clinical trials targeting ncRNAs for the treatment of AML? If not, what is the major issues to prevent them to be used as targets? Have any ncRNAs been validated in large patient cohorts or included in diagnostic panels for AML?
- Figures could be more informative. For examples:
- Figure 1 could include six WHO classification of AML with major clinical characteristics as summarized in Table 1.
- Figure 2 could include more detailed information on specific ncRNAs in each listed progress of AML on cancer stem cells, cell cycle, proliferation, differentiation, development, and apoptosis. The current Figure 2 is too general without any meaningful information for AML. The figure can be true for any cancer. The figure legend is too simple without any guidance information to help to understand the information in the figure.
- Figure 3 has same problem without any specific information for the roles of ncRNAs in the genes important for AML. Some examples for how a specific lncRNA or circRNA interacts with miRNAs in control critical genes in the progress of AML could be more informative.
Author Response
This manuscript provides a comprehensive review of the current literature on the roles of noncoding RNAs (ncRNAs, including miRNAs, lncRNAs, and circRNAs) in the complex landscape of acute myeloid leukemia (AML). The review covers an important and timely topic on cancer biology with a focus on ncRNAs, which is fitted well with the scopes of the journal. However, the current version of the manuscript requires significant revision to enhance its analytical depth, clinical relevance, and clarity. Addressing the points above would greatly improve the manuscript’s quality and impact.
Response: We thank the reviewer for this valuable suggestion. As suggested by the respected reviewer the manuscript has been carefully revised incorporating all suggested modifications given by respected reviewer. Point wise response to all comments given below.
Major concerns :
Comment 1: The manuscript lacks critical discussion in some sections. While many ncRNAs are listed, the review does not sufficiently analyze or compare their relative strengths or clinical utility. A more analytical, less descriptive approach is recommended.
Response: We are very much thankful to the reviewer for this important observation. The revised manuscript now includes a more a new secion “ LncRNAs : Emriging Clinical Utility” , in this section we discussed clinical relevance of different ncRNAs, particularly in the context of their diagnostic, prognostic, and therapeutic potential. These revisions provide a more critical and comparative perspective, addressing the respected reviewer’s concern and strengthening the manuscript’s scientific value.
Comment 2: The section of “Mechanisms of ncRNAs in AML” needs to be expanded with better depth on the analysis of how ncRNAs involved in the more biological insight in the development of AML. More mechanistic analysis is required.
Response: We appreciate the reviewer’s constructive suggestion. The section “Mechanisms of ncRNAs in AML” has been significantly expanded and subdived to provide better clarity and deeper mechanistic insights. These revisions add greater analytical depth and enhance the mechanistic understanding of ncRNAs in AML pathogenesis.
Comment 3: The manuscript should more clearly address how lncRNAs are being translated into clinical applications (add a new section). Are there any ongoing clinical trials targeting ncRNAs for the treatment of AML? If not, what are the major issues to prevent them to be used as targets? Have any ncRNAs been validated in large patient cohorts or included in diagnostic panels for AML?
Response: We are really thankful to the reviewer for this valuable suggestion. A new section has been added to the manuscript titled “LncRNAs: Emrging Clinical Utility.” (highlighted yellow) This section discusses the current status of lncRNAs as biomarkers and therapeutic targets, with emphasis on their potential in diagnosis, prognosis, and treatment strategies. These revisions provide a clearer picture of the translational potential and current limitations of lncRNA research in AML.
Comment 4: Figures could be more informative. For examples:
-
Figure 1 could include six WHO classification of AML with major clinical characteristics as summarized in Table 1.
-
Figure 2 could include more detailed information on specific ncRNAs in each listed progress of AML on cancer stem cells, cell cycle, proliferation, differentiation, development, and apoptosis. The current Figure 2 is too general without any meaningful information for AML. The figure can be true for any cancer. The figure legend is too simple without any guidance information to help to understand the information in the figure.
-
Figure 3 has same problem without any specific information for the roles of ncRNAs in the genes important for AML. Some examples for how a specific lncRNA or circRNA interacts with miRNAs in control critical genes in the progress of AML could be more informative.
Response:
-
We appreciate the reviewer for these constructive suggestions regarding our figures.
-
Figure 1 has been revised for better clarity. This figure is to show the how AML begins. Adding 6 different class of AML may mislead the manuscript theme. I respect reviewer’s suggestion but graphical representation of 6 classes of AML not well suited here.
-
Figure 2 has been expanded to provide detailed information on the roles of specific ncRNAs in AML processes. The figure legend has also been rewritten to offer clearer guidance for interpretation.
-
We appreciate the reviewer’s valuable suggestion. As alreday discussed that for better clarity Figure 3 has been removed now.
-
Together, these revisions significantly improve the informativeness and clarity of the figures, making them directly relevant and valuable for understanding the role of ncRNAs in AML.

Round 2
Reviewer 1 Report
Comments and Suggestions for Authors
There remain just some minor formatting (Table 1) to produce publication-quality Figures and Tables. I am pleased with the revisions made and the attention to detail. I am particularly grateful for the updated Figure 2. There may be a formatting error in the top left blue circle which lists TUG1 and MEG3 where a process is expected. The manuscript is greatly improved in clarity, readability, content, and consistency.
Author Response
We are thankful for your positive remarks on the improved clarity, readability, and consistency of the manuscript. All the suggested modifications incorporated in revised manuscript. We carefully revised Table 1 to ensure proper formatting for publication quality. Figure 2 has now been rectified to clearly show the expected process.
Reviewer 2 Report
Comments and Suggestions for Authors
The authors have adequately addressed my concerns.
Author Response
We thank respected reviewer for confirming that all concerns have been addressed.
Reviewer 3 Report
Comments and Suggestions for Authors
The authors have addressed my comments in an appropriate manner. I don't have any further concern.
Author Response
We sincerely thank respected reviewer for acknowledging that your concerns have been adequately addressed.